# Lipid-Lowering Potential of Almond Hulls (Quercetin, Baicalein, and Kaempferol): Insights from Network Pharmacology and Molecular Dynamics

**DOI:** 10.3390/cimb47060450

**Published:** 2025-06-12

**Authors:** Qiming Miao, Lu Sun, Jiayuan Wu, Xinyue Zhu, Juer Liu, Roger Ruan, Guangwei Huang, Shengquan Mi, Yanling Cheng

**Affiliations:** 1Biochemical Engineering College, Beijing Union University, Beijing 100101, China; 2Center for Biorefining & Department of Bioproducts and Biosystems Engineering, University of Minnesota, St. Paul, MN 55108, USA; 3Almond Board of California, Modesto, CA 95354, USA

**Keywords:** almond hull, extraction, mechanisms, HepG2 cells

## Abstract

The advancement of modern lifestyles has precipitated excessive consumption of energy-dense foods, driving the escalating global burden of lipid metabolism dysregulation-related pathologies—including obesity, type 2 diabetes mellitus (T2DM), non-alcoholic fatty liver disease (NAFLD), and cardiovascular disorders—which collectively pose a formidable challenge to global public health systems. The almond hull, as a by-product of almond processing, is rich in polyphenolic compounds with demonstrated antioxidant, anti-inflammatory, and lipid-lowering potential, though its precise hypo-lipidemic mechanisms remain elusive. In this study, polyphenols were extracted from almond hulls using 50% ethanol with ultrasound-assisted extraction, followed by preliminary purification via solvent partitioning. The ethyl acetate fraction was analyzed by liquid chromatography–mass spectrometry (LC-MS). Network pharmacology and molecular docking were employed to investigate the interactions between key bioactive constituents (e.g., quercetin, baicalein, and kaempferol) and targets in lipid metabolism-related pathways. Molecular dynamics (MD) simulations further evaluated the stability of the lowest-energy complexes. Results revealed that the ethyl acetate fraction exhibited potent pancreatic lipase inhibitory activity (IC50 = 204.2 µg/mL). At 0.1 mg/mL after 24 h treatment, it significantly reduced free fatty acids (FFAs)-induced intracellular triglyceride accumulation (*p* < 0.01) and enhanced cellular antioxidant capacity. Network pharmacology and in vitro studies suggest almond hull extract modulates PI3K-AKT signaling and improves insulin resistance, demonstrating lipid-lowering effects. These findings support its potential in functional foods and pharmaceuticals, though further in vivo validation and mechanistic investigations are required.

## 1. Introduction

Lipid metabolism constitutes a fundamental physiological process governing energy homeostasis, cellular membrane integrity, and signal transduction through its involvement in energy storage, hormonal modulation, neurosynaptic transmission, and transport of lipophilic nutrients [1]. Dysregulation of this system manifests pathologically as elevated plasma cholesterol, hypertriglyceridemia, and lipoprotein dysfunction, driving pathogenesis in atherosclerosis, type 2 diabetes mellitus (T2DM), non-alcoholic fatty liver disease (NAFLD), and obesity [2]. Epidemiological projections highlight a global surge in metabolic disorders [3], with 750 million pediatric and 3.3 billion adult cases of overweight/obesity anticipated by 2035. Concurrently, the global diabetic population is predicted to reach 783 million by 2045 [4,5].

Mechanistic studies reveal multifactorial interactions underpinning lipid dysregulation, including aberrant transcriptional regulation (e.g., SREBP/PCSK9 axis dysfunction), chronic inflammation mediated through TNF-α/mTORC1 signaling, and microbiota–host crosstalk via short-chain fatty acid metabolism [6]. Phytochemicals have emerged as promising therapeutic agents for modulating glucolipid metabolism and insulin sensitivity, demonstrating efficacy in preclinical models of metabolic syndrome [7].

In almond (*Amygdalus communis* L.), the hull constitutes approximately 52% of the total fresh weight, while the kernel with integument represents merely 15% [8,9]. However, the increasing utilization of almond kernels has generated substantial by-products, particularly hulls, necessitating sustainable valorization strategies. For instance, California’s almond industry produced ~1.85 billion kg of hulls and 753 million kg of shells in 2022 [10], with global annual hull output exceeding 6 million metric tons [11]. Currently, the predominant disposal methods for almond hulls remain limited to incineration and livestock feed applications, resulting in suboptimal resource utilization. Conducting systematic investigations into the valorization of almond hulls holds substantial promise for advancing circular economy principles and generating enhanced economic returns. The almond hull primarily comprises sugar, crude fiber, and proteins, while also containing functional bioactive constituents dominated by triterpenoids, lactones, and phenolic compounds [12], including oleanolic acid [13], ursolic acid [14], catechin derivatives [15], flavonol aglycones [16], and glycosylated flavanones [17].

As ubiquitous phytochemicals and secondary metabolites, plant polyphenols are predominantly localized in floral, seed, and root tissues of edible botanicals [18]. Substantial evidence confirms their multidimensional bioactivity encompassing antimicrobial/viral suppression and therapeutic potential in combating neoplasia, cardiovascular pathologies, neurodegenerative disorders, obesity, and diabetes mellitus [19]. Valorizing this by-product into functional ingredients or dietary supplements has emerged as a research priority. Notably, 8% hull powder incorporation in bread formulations elevates dietary fiber content 2.3-fold while achieving 60.5% polyphenol bioavailability [20], though systematic investigations into physicochemical modification, processing compatibility, and consumer acceptance remain imperative. Harnafi et al. demonstrated that almond hull phenolic extracts ameliorate lipid dysregulation in Triton WR-1339/hyperlipidemic diet-induced murine models [21,22], yet mechanistic elucidations are conspicuously absent. Therefore, systematic investigation of the functional properties of almond hull polyphenols provides critical insights for facilitating valorization strategies, promoting circular bioeconomy frameworks, and enhancing economic viability within agro-industrial sectors. Network pharmacology provides a robust framework for deciphering the poly-pharmacological mechanisms underlying multi-component synergy, and multi-pathway interventions in natural products. In molecular interaction studies, computational tools such as molecular docking analyze stereospecific binding modalities between small-molecule ligands, which are macromolecular targets [23]. Complementarily, molecular dynamics simulations quantify binding free energy and the affinity to predict ligand–receptor conformational stability [24,25].

Combing network pharmacology with docking and molecular dynamics simulations provides an effective strategy to investigate the lipid metabolism-modulating components in almond hull extracts. This research employs network pharmacology to map bioactive constituents and their target pathways, with subsequent experimental validation of in vitro levels.

## 2. Materials and Methods

### 2.1. Materials

Almond hulls were supplied by the Almond Board of California; HepG2 cells were sourced from Peking Union Medical College, Institute of Basic Medical Sciences; CCK-8 assay kit was from Yeasen Biotechnology, Shanghai, China; DMEM high-glucose medium and Fetal bovine serum and EDTA-trypsin were sourced from Procell Life Science, Wuhan, China; sodium oleate and palmitate and pancreatic lipase were from Aladdin Biochemical, Shanghai, China; ethanol (analytical grade), ethyl acetate, and petroleum ether were from Beijing Zhengcheng Biotechnology Co., Ltd., Beijing, China. CO_2_ incubator was from BSC-1360LIIA2; SANYO, Osaka, Japan; multi-mode microplate reader was from INFINITE M NANO; Thermo Fisher Scientific, Waltham, MA, USA; ultracentrifuge was from 3K15; Thermo Fisher Scientific, Waltham, MA, USA; inverted microscope was supplied by Nikon Eclipse TS2; Nikon, Tokyo, Japan.

### 2.2. Extraction of Bioactive Compounds from Almond Hulls

Following sequential pretreatment (decontamination, desiccation, and pulverization through a 100-mesh sieve), the almond hull powder underwent ultrasonic-assisted extraction (UAE) employing 50% ethanol (1:20 solid-to-solvent ratio) under 270 W irradiation for 30 min. The resultant ethanolic extract was concentrated via rotary evaporation prior to polarity-guided fractionation. Initial triple liquid–liquid partitioning with petroleum ether (1:1 *v*/*v*, 30 min per cycle) yielded lipophilic fractions, while the residual aqueous phase subsequently underwent triplicate ethyl acetate partitioning (1:1 *v*/*v*, 60 min per cycle) to isolate semi-polar constituents. This multi-step fractionation protocol generated four chemically distinct phases: ethanolic extract concentrate, petroleum ether-soluble fraction, ethyl acetate-soluble fraction (AHE), and residual aqueous fraction.

### 2.3. Network Pharmacology, Molecular Docking, and Molecular Dynamics Simulation

#### 2.3.1. Bioactive Compound Screening

The LC-MS-derived phytochemical profile of the almond hull ethyl acetate-soluble fraction was systematically interrogated using the TCMSP database (https://www.91tcmsp.com/#/database (accessed on 29 August 2024)) to screen potential target active compounds according to the criteria of oral bioavailability (OB) ≥ 30% and drug-likeness (DL) ≥ 0.18. The chromatographic separation was performed on an Agilent SB-C18 column (1.8 µm particle size, 2.1 mm × 100 mm; Agilent, Santa Clara, CA, USA) with a mobile phase consisting of 0.1% formic acid in water (phase A) and acetonitrile supplemented with 0.1% formic acid (phase B). The flow rate was maintained at 0.35 mL/min, with a column temperature of 40 °C and an injection volume of 2 µL. Analytical data were acquired using an ultra-high-performance liquid chromatography system coupled to a tandem mass spectrometer. The gradient elution conditions and mass spectrometry parameters are listed in Table 1 and Table 2.

#### 2.3.2. Computational Screening and Prediction of Bioactive Compound–Disease Target Interactions

The potential targets of bioactive compounds were systematically predicted through in silico approaches. Initially, SMILES notations of the compounds were submitted to multiple databases, including TCMSP (https://www.91tcmsp.com/#/database (accessed on 29 August 2024)), ETCM (http://www.tcmip.cn/ETCM (accessed on 29 August 2024)), Therapeutic Target Database (TTD, https://db.idrblab.net (accessed on 29 August 2024)), and BATMAN-TCM (http://bionet.ncpsb.org/batman-tcm/ (accessed on 29 August 2024)) with stringent parameters (Confidence Score cutoff: 0.84, LR = 80.88, *p*-value < 0.05, Druggability Score ≥ 0.1). PubChem (https://www.ncbi.nlm.nih.gov/ (accessed on 29 August 2024)) was concurrently accessed to verify chemical structures. Subsequently, reverse target fishing analysis was performed via the Swiss Target Prediction platform (http://www.swisstargetprediction.ch (accessed on 30 August 2024)) to identify candidate targets.

For disease target mapping, obesity-related genes were retrieved by querying the keyword “Obesity” in GeneCards (https://www.genecards.org (accessed on 6 September 2024)) and DisGeNET (https://www.disgenet.org (accessed on 6 September 2024)), followed by data consolidation and normalization to eliminate redundancies.

#### 2.3.3. Retrieval of Overlapping Genes Between Bioactive Compounds and Diseases

To identify overlapping targets between bioactive compounds and diseases, the screened target datasets (including bioactive compound-related targets and disease-associated targets) were imported into the Draw Venn Diagram tool (available at: http://bioinformatics.psb.ugent.be/webtools/Venn/ (accessed on 8 September 2024)). This analysis directly generated the intersection of targets shared between bioactive compounds and diseases.

#### 2.3.4. Network Construction and Therapeutic Target Identification in Protein–Protein Interaction Analysis

The common genes were uploaded to the STRING database (version 3.5, https://cn.string-db.org/ (accessed on 8 September 2024)) to generate a primary protein–protein interaction (PPI) network. The resulting PPI data were imported into Cytoscape (version 17.3.8.0) for topological characterization using three centrality metrics: degree, closeness centrality (CC), and between-ness centrality (BC). Targets exceeding the median values of these metrics were identified as core nodes, while bioactive components associated with the highest number of targets were prioritized as key constituents. Potential disease-related chemical compounds and critical targets were subsequently derived from this analysis. A protein–target interaction network was then constructed in Cytoscape, integrating these core targets and bioactive components to visualize their interplay.

#### 2.3.5. KEGG and GO Enrichment Analysis

Gene set enrichment analysis was performed using Metascape (version 3.5, http://metascape.org/gp/ (accessed on 9 September 2024)) to annotate biological functions of prioritized targets. The intersection gene list was submitted to the platform with Homo sapiens selected as the species parameter. Functional annotations were systematically explored through Gene Ontology (GO) and Kyoto Encyclopedia of Genes and Genomes (KEGG) pathway analyses under the Enrichment module. The analytical report, generated via the Enrichment Analysis tab, provided comprehensive insights into biological processes, molecular functions, and disease-related pathways. For visualization, we used http://www.bioinformatics.com.cn/ (accessed on 9 September 2024) to generate publication-quality graphical representations of enrichment results.

#### 2.3.6. Molecular Docking

The identified bioactive compounds were retrieved in mol2 format from the PubChem database (https://pubchem.ncbi.nlm.nih.gov/ (accessed on 10 October 2024)). Core target proteins were acquired by inputting gene names into the UniProt platform (https://www.uniprot.org/ (accessed on 10 October 2024)). Protein receptors were preprocessed using AutoDockTools 1.5.6, including hydrogenation and parameter configuration, and exported as PDBQT files. Molecular docking was performed with AutoDock Vina, and binding conformations were visualized using PyMOL (Version 2.5.7).

#### 2.3.7. Molecular Dynamics (MD) Simulations

Molecular dynamics (MD) simulations were performed using Gromacs 12.3 on the receptor–ligand complex exhibiting the highest binding affinity. The initial structure, derived from molecular docking results, underwent energy minimization followed by a 30 ns simulation under standard temperature and pressure (STP) conditions. Simulations utilized a 100 ps time-step and default parameters for other settings.

### 2.4. In Vitro Lipid-Lowering Assay

#### 2.4.1. Pancreatic Lipase Inhibition Assay

A pancreatic lipase activity assay was conducted using p-nitrophenyl butyrate (PNPB) as the substrate [26]. Hydrolysis of PNPB by the enzyme releases using p-nitrophenol, which exhibits a yellow chromophore under alkaline conditions. Enzymatic inhibition was quantified by measuring absorbance at 405 nm, proportional to residual substrate hydrolysis.

The reaction system is detailed in Table 3, in 96-well plates, 50 µL of pancreatic lipase solution was mixed with 20 µL of test samples (ethanol extract, ethyl acetate fraction, or aqueous fraction) dissolved in DMSO-containing buffer at concentrations of 0.0625, 0.125, 0.25, 0.5, and 1 mg/mL. After adding 110 µL Tris-HCl buffer (pH 8.0), the mixture was pre-incubated at 37 °C for 15 min. Reactions were initiated with 20 µL PNPB and incubated for 30 min under identical conditions before absorbance measurement.$$Enzymatic\;inhibition\;activity\;(\%)=(1-\frac{\mathrm{Sample}\;\mathrm{group}\;\mathrm{absorbance}-\mathrm{sample}\;\mathrm{control}\;\mathrm{group}}{\mathrm{Blank}\;\mathrm{group}\;\mathrm{absorbance}-\mathrm{Blank}\;\mathrm{control}\;\mathrm{group}})\times 100$$

#### 2.4.2. Cellular Lipid-Lowering Experiment

##### Cell Survival Assay

HepG2 cells were seeded in 96-well plates at 1 × 10^4^ cells/well (100 µL/well) and allowed to adhere. Subsequently, equal volumes of AHE working solution (concentrations: 0.2, 0.4, 0.6, 0.8, 1 mg/mL) and FFA working solution (concentrations: 0.1, 0.2, 0.4, 0.5, 0.6 mM/L) were added to designated wells, followed by incubation for 24 or 48 h. After treatment, cells were incubated with CCK-8 reagent (10% *v*/*v* in basal medium) for 30 min. Absorbance was measured at 450 nm to calculate cell viability.

##### Quantitative Analysis of Cellular Triglycerides, Total Cholesterol, and Antioxidant Biomarkers

HepG2 cells in logarithmic growth phase (4 × 10^5^ cells/well) were seeded into 6-well plates and induced with an FFA mixture (oleate/palmitate, 2:1 ratio, 0.5 mM) to establish a lipid accumulation model. Cells were co-treated with varying concentrations of Artemisia herba-alba extract (AHE: 0.1, 0.2, and 0.4 mg/mL) and categorized into five groups: N (normal control, untreated), M (FFA-induced model), MEL (M + 0.1 mg/mL AHE), MEM (M + 0.2 mg/mL AHE), and MEH (M + 0.4 mg/mL AHE). Post-incubation, cellular triglyceride (TG), total cholesterol (T-CHO), glutathione (GSH), malondialdehyde (MDA), and superoxide dismutase (SOD) levels were quantified using commercial assay kits.

##### Oil Red O Staining

HepG2 cells in the logarithmic growth phase were seeded into 6-well plates at a density of 4 × 10^5^ cells/well. After 24 h incubation, cells were fixed with 4% (*w*/*v*) paraformaldehyde at room temperature for 30 min, subsequently stained with freshly prepared Oil Red O solution under light-protected conditions for 30 min, and rinsed with 60% isopropanol. Lipid droplet morphology was visualized and documented using fluorescence microscopy.

### 2.5. Statistical Methods

Statistical analyses were performed using IBM SPSS Statistics 26 (Armonk, NY, USA). Continuous data are expressed as mean ± standard deviation (SD), and one-way ANOVA was applied for group comparisons. Graphical representations were generated with Origin 2021 (Northampton, MA, USA) and GraphPad Prism 8 (San Diego, CA, USA). Statistical significance was defined as *p* < 0.05, and *p* < 0.01 was considered highly significant. All experiments, including pancreatic lipase inhibition and cellular assays, were conducted with *n* ≥ 3 technical replicates per group.

## 3. Results

### 3.1. Network Pharmacology Analysis

#### 3.1.1. TCMSP-Driven Screening and Identification of Potential Bioactive Compounds

Based on LC-MS analysis, the phytochemicals identified in our study were subjected to systematic screening using the TCMSP database with established pharmacokinetic parameters: oral bioavailability (OB) ≥ 30% and drug-likeness (DL) ≥ 0.18. This rigorous screening process yielded 19 potential bioactive compounds exhibiting favorable pharmacokinetic profiles (Table 4), with OB values ranging from 30.68% to 56.55% and DL scores between 0.21 and 0.84.

#### 3.1.2. Intersection Analysis of Bioactive Compounds and Disease Targets

The 19 bioactive compounds were mapped to 821 potential targets using compound–target interaction databases. To identify obesity-associated therapeutic targets, “obesity” was queried in disease-specific databases, yielding 2456 disease-related targets. Intersection analysis revealed 298 overlapping targets between the bioactive compounds and obesity pathology, as depicted in Figure 1.

#### 3.1.3. Protein–Protein Interaction Network

The intersected genes were analyzed using the STRING 12.0 database with high-confidence filtering (score > 0.4), generating a primary protein–protein interaction (PPI) network (Figure 2a). The raw network file was subsequently imported into Cytoscape 3.10.0 for topological refinement, where nodes were ranked by centrality metrics including degree, between-ness centrality (BC), and closeness centrality (CC). In the refined network (Figure 2b), node importance was proportional to centrality values, with color intensity reflecting degree magnitude and edge density indicating interaction robustness. Hub analysis identified GAPDH, IL6, AKT1, TNF, ACTB, IL1B, TP53, and STAT3 as top-ranked nodes within the network core, demonstrating extensive interactivity.

#### 3.1.4. GO and KEGG Analysis

Intersection genes were analyzed in Metascape 3.5 (species: Homo sapiens) for functional enrichment. Gene Ontology (GO) terms—including biological processes (BP), cellular components (CC), and molecular functions (MF)—were prioritized based on the top 10 entries ranked by *p*-value for each category. KEGG pathway enrichment identified the top 20 significant pathways (*p* < 0.05). Visualization was performed using a bioinformatics platform to generate interpretable outputs.

As shown in Figure 3, GO enrichment analysis revealed significant associations with biological processes (BP) such as hormone response, cellular lipid regulation, and exogenous stimulus adaptation. Cellular components (CC) were predominantly enriched in membrane microdomains (e.g., rafts and lateral regions), while molecular functions (MF) clustered around DNA–transcription factor binding and translational regulation. KEGG pathway analysis (Figure 4) identified key targets linked to lipid metabolism and atherosclerosis, cancer progression, AGE-RAGE signaling in diabetes, fluid shear stress, chemical carcinogen–receptor activation, and PI3K-AKT signaling. Integrative analysis highlighted lipid-related mechanisms (cellular lipid response, PI3K-AKT signaling, insulin resistance, and non-alcoholic fatty liver disease) and inflammation-associated pathways (TNF/IL-17 signaling). These findings suggest that the bioactive components of almond hulls modulate disease phenotypes through multi-target and multi-pathway interactions.

#### 3.1.5. Compound–Target–Disease Network

As illustrated in Figure 5, a network integrating common targets, bioactive components (quercetin, baicalein, and kaempferol), and KEGG pathways was constructed using Cytoscape. These core compounds exhibited distinct target profiles (quercetin: 208 targets; baicalein: 103; kaempferol: 94). Molecular docking was subsequently performed to validate interactions between these components and key targets within lipid metabolism-related pathways (e.g., PI3K-AKT signaling, insulin resistance).

#### 3.1.6. Molecular Docking and Molecular Dynamic Simulation

As summarized in Table 5 and Figure 6, key targets from the KEGG lipid metabolism-related pathways (PI3K-AKT signaling and insulin resistance) predicted by network pharmacology were prioritized for molecular docking. Panels a–f in Figure 6 visualize the binding modes of quercetin, baicalein, and kaempferol with critical targets (mTOR, PI3K, AKT1, STAT3, TNF, and IL6). The calculated binding energies for these ligand–target pairs were consistently below −5 kcal/mol (Table 3), indicating robust molecular interactions. These findings suggest that the identified components and their targets within the PI3K-AKT and insulin resistance pathways may play pivotal roles in ameliorating lipid dysregulation in hyperlipidemic models. Furthermore, the lipid-lowering mechanism of almond exocarp likely involves modulation of these pathways.

To validate ligand–target binding stability, molecular dynamics (MD) simulations were performed on the mTOR complexes with baicalein, kaempferol, and quercetin—selected based on optimal docking energies. Radius of gyration (Rg) analysis (Figure 7a) revealed stable compactness of all complexes within the initial 20 ns. Post-20 ns, mTOR–quercetin exhibited increased Rg, suggesting structural loosening due to altered protein rigidity. Root-mean-square deviation (RMSD) and fluctuation (RMSF) metrics [27] further quantified conformational stability. mTOR–baicalein demonstrated minimal RMSD fluctuations (0.375–3.083 Å), while mTOR–kaempferol maintained <5.121 Å RMSD with post-10 ns variations below 2 Å (Figure 7b). In contrast, mTOR–quercetin showed pronounced RMSD shifts, indicating reduced stability. RMSF profiling (Figure 7c) highlighted conserved residue-specific flexibility (217 residues analyzed) across mTOR–baicalein and mTOR–kaempferol complexes, aligning with unbound mTOR. Conversely, quercetin induced atypical fluctuations, implying distinct binding interactions. Collectively, MD simulations corroborated the docking results, confirming baicalein and kaempferol as stable modulators of mTOR with potential lipid-regulatory relevance.

### 3.2. In Vitro Evaluation of Hypolipidemic Efficacy

#### 3.2.1. Lipase Inhibitory Effects

As shown in Figure 8, the ethyl acetate fraction exhibited a concentration-dependent inhibitory effect on lipase activity, whereas both ethanol extract and aqueous phase displayed nonlinear dose–response relationships characterized by initial inhibitory enhancement, followed by attenuation at elevated concentrations. Notably, the ethyl acetate fraction derived from almond hull ethanol extract demonstrated superior lipase inhibitory efficacy, with determined IC50 values of 204.2 µg (ethyl acetate layer), 323.3 µg (ethanolic extract), and 13,727 µg (aqueous phase). In vitro porcine pancreatic lipase inhibition assays confirmed the enhanced lipid-reducing capacity of the ethyl acetate-soluble constituents, as evidenced by its significantly lower IC50 compared to other fractions. These findings suggest differential distribution of bioactive compounds across extraction phases, with the ethyl acetate fraction containing the most potent inhibitors of lipid-digestive enzymes. The parabolic response patterns observed in polar fractions may reflect competitive inhibition mechanisms or phytochemical instability at supraphysiological concentrations.

#### 3.2.2. Cell-Based Investigation Results

##### Cell Proliferation Assay Data Interpretation

The ethyl acetate layer demonstrated potent pancreatic lipase inhibitory activity in vitro, prompting further evaluation of its lipid-modulating effects in cellular models. To establish a cellular hypertriglyceridemia model and determine the effective concentration of the extract, we conducted cell viability analyses on both FFA modeling concentrations and extract concentrations. Quantitative analysis (Figure 9a) revealed concentration-dependent attenuation of cellular viability following 24 h exposure, with statistically significant reductions observed at 0.6 mg/mL (*p* < 0.05) compared to vehicle controls. Enhanced inhibitory efficacy was documented at elevated concentrations (0.8–1 mg/mL, *p* < 0.01). Extended exposure duration (48 h, Figure 9b) intensified these pharmacological effects, particularly at 0.6 mg/mL. Cytotoxicity assessment established 0.4 mg/mL as the non-toxic threshold for 24 h interventions.

FFA-induced lipo-toxicity manifested marked cytotoxicity at 0.6 mg/mL (24 h, *p* < 0.05), whereas extended exposure (48 h) resulted in significant viability impairment at both 0.5 mg/mL and 0.6 mg/mL (*p* < 0.05) (Figure 9c,d). Notably, the 24 h time frame exhibited a concentration-dependent toxicity threshold, justifying the selection of 0.5 mg/mL as the optimal induction dose to balance model validity and minimize prolonged cytotoxic effects. Based on these pharmaco-dynamic profiles, experimental parameters were standardized as follows: 24 h exposure to 0.5 mg/mL FFAs for lipo-toxicity modeling, coupled with co-treatment using the fraction at sub-toxic concentrations (≤0.4 mg/mL) for equivalent duration.

##### Intracellular Triglyceride and Total Cholesterol

Figure 10a demonstrated statistically significant divergence between the lipo-toxic model (Group M) and normo-lipidemic controls (Group N) (*p* < 0.01), confirming successful establishment of the pathological model. Dose-dependent attenuation of lipid accumulation was observed in extract-treated groups (MEL, MEM, MEH), with all concentrations showing marked reduction compared to Group M (*p* < 0.01). Figure 10b revealed no significant cholesterol elevation in 0.5 mM FFA-exposed cells versus controls (*p* > 0.05). While MEH treatment exhibited a downward trend in cholesterol content relative to Group M, this alteration did not reach statistical significance compared to either model or control groups (*p* > 0.05). Cellular analyses demonstrated concentration-dependent attenuation of triglyceride accumulation by the extract, with high-dose groups showing marginal cholesterol reduction.

##### Oil Red O Staining Result

As evidenced by Oil Red O staining in Figure 11, distinct morphological differences were observed between experimental groups. The M group exhibited intensified red staining with expanded lipid droplet areas compared to N group, confirming successful induction of lipidosis by 0.5 mM FFAs. Comparative analysis revealed dose-dependent mitigation of neutral lipid deposits in extract-treated groups (MEL/MEM/MEH). Oil Red O staining confirmed successful establishment of the lipid accumulation model through 0.5 mM FFAs induction, with the extract demonstrating significant attenuation of FFA-induced intracellular lipid deposition.

##### Results of Cellular Antioxidant Indicators

As shown in Figure 12a, superoxide dismutase (SOD) activity in the model group (M) decreased significantly versus the N group (*p* < 0.05). All extract-treated groups exhibited dose-dependent SOD restoration, with medium- and high-dose groups (ME-M/ME-H) demonstrating statistically significant enhancements compared to the M group (*p* < 0.01), while the ME-L showed non-significant elevation. Figure 12b illustrates malondialdehyde (MDA) levels across groups. The M group displayed an MDA accumulation relative to the N group (*p* < 0.05). Extract intervention reduced MDA content in all treatment groups, though only ME-M achieved statistical significance versus the M group (*p* < 0.05), whereas ME-L and ME-H groups exhibited non-significant mitigation. For glutathione (GSH) quantification (Figure 12c), the M group showed a non-significant reduction compared to the N group (*p* > 0.05). Extract administration induced GSH recovery across treatment groups, but these changes did not reach statistical significance.

## 4. Discussion

Obesity, a primary manifestation of lipid metabolism disorders, is closely associated with insulin resistance, dyslipidemia, impaired glucose metabolism, and hypertension, particularly in cases of visceral adiposity [28]. As a promising yet underutilized by-product, almond hulls have demonstrated lipid-lowering properties, though their mechanistic underpinnings remain underexplored. This study employed network pharmacology and molecular docking to systematically analyze bioactive components in almond hulls, identifying critical phytochemicals and potential mechanisms subsequently validated through in vitro experiments. Three polyphenolic compounds—baicalein, kaempferol, and quercetin—were identified as key active constituents. Baicalein exhibits multifaceted pharmacological activities, including antioxidative, anti-inflammatory, and antidiabetic effects, with mechanisms involving suppression of carbohydrate hydrolases, inhibition of adipogenesis, and modulation of adipocyte differentiation [29,30,31]. Kaempferol regulates insulin signaling pathways, ameliorates insulin resistance, and enhances glucose metabolism through interactions with NF-κB, SIRT1, and AMPK cascades [32,33,34]. Quercetin, a flavonoid derivative, demonstrates therapeutic potential against metabolic syndrome via lipid-lowering and anti-inflammatory actions [35].

KEGG enrichment analysis revealed the PI3K-AKT signaling pathway and insulin resistance pathway as central to lipid metabolism regulation. The PI3K-AKT axis regulates hepatic lipogenesis, gluconeogenesis, and glucose uptake, while concurrently modulating adipogenesis through insulin sensitization [36,37,38,39]. As pivotal lipid kinases, PI3Ks generate secondary messengers (PI(3)P, PI(3,4)P2, PI(3,4,5)P3) that orchestrate cellular biosynthesis, survival, and metabolism [40,41,42]. AKT, the nodal kinase in this cascade, governs lipoprotein lipase (LPL) activity. PI3K/AKT/mTOR inhibition exacerbates hyperlipidemia via impaired lipid clearance [43,44]. Notably, mTOR complex 1 (mTORC1) integrates nutrient/growth factor signals to regulate anabolic processes [45,46,47]. Hepatic steatosis epidemiologically correlates with insulin resistance [48,49,50,51], wherein defective PI3K/AKT-mediated suppression of lipolysis elevates circulating FFAs and TG [52,53,54]. Compensatory activation of sterol regulatory element-binding protein 1c (SREBP-1c) sustains de novo lipogenesis despite insulin resistance [55], while impaired adipocyte insulin response exacerbates dyslipidemia through reduced FFA uptake and TG synthesis [56,57,58].

These findings suggest that almond hulls may attenuate lipid metabolic dysregulation through coordinated modulation of the PI3K-AKT/mTOR pathway and insulin resistance signaling, offering a mechanistic basis for their traditional use in lipid management. Further validation of target interactions (e.g., mTOR, STAT3, TNF) is warranted to elucidate precise molecular cascades.

## 5. Conclusions

This study comprehensively elucidated the hypolipidemic mechanisms of almond hull extract through an integrated approach combining network pharmacology, molecular docking, dynamic simulations, and in vitro validation. Key findings demonstrated significant lipid-lowering effects through pancreatic lipase inhibition (in vitro assay) and reduced intracellular triglyceride accumulation (HepG2 cell model), accompanied by enhanced antioxidant capacity. Our results not only confirm the extract’s anti-obesity potential but also identify critical signaling pathways for future mechanistic validation. These findings establish a theoretical foundation for developing almond hull-derived nutraceuticals targeting metabolic syndrome. Future studies will focus on compound purification and in vivo/in vitro mechanistic investigations to further validate these pathways.

## Figures and Tables

**Figure 1 cimb-47-00450-f001:**
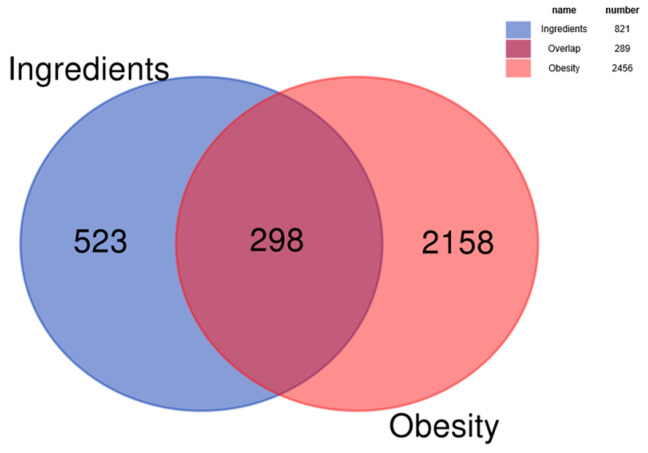
Venn diagram of the intersection of disease and ingredient compound targets.

**Figure 2 cimb-47-00450-f002:**
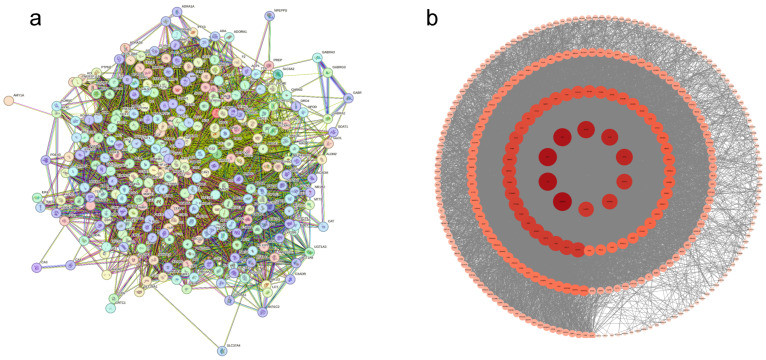
Intersection target PPI (protein–protein interaction) network diagram. (**a**) Original protein–protein interaction. (**b**) Protein–protein interaction.

**Figure 3 cimb-47-00450-f003:**
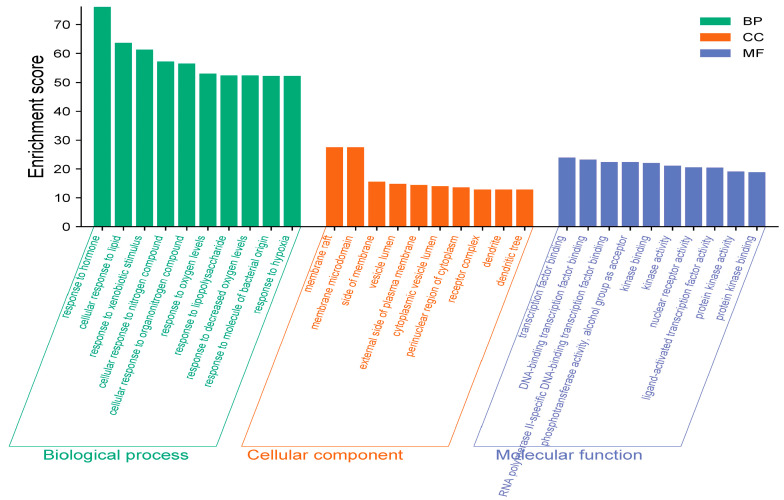
GO enrichment analysis chart.

**Figure 4 cimb-47-00450-f004:**
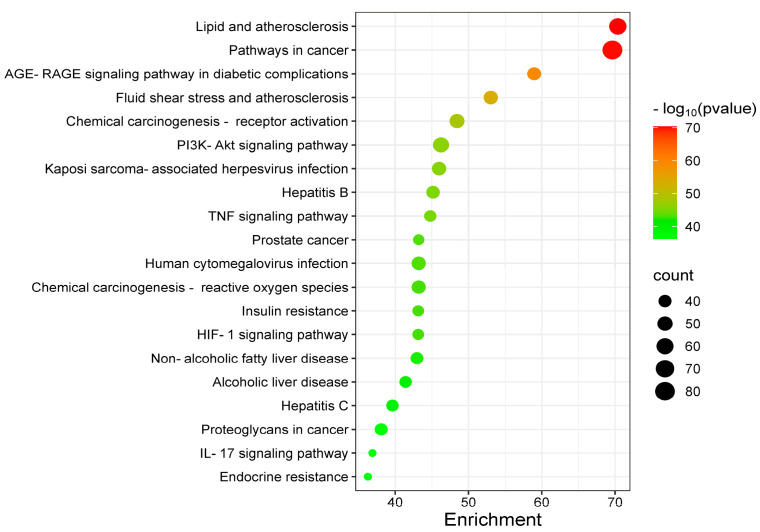
KEGG enrichment analysis chart.

**Figure 5 cimb-47-00450-f005:**
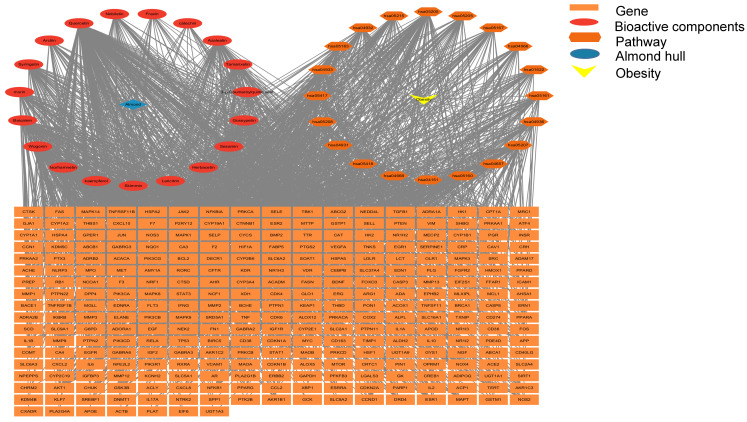
Diagram of the network relationship between components, potential targets, and diseases.

**Figure 6 cimb-47-00450-f006:**
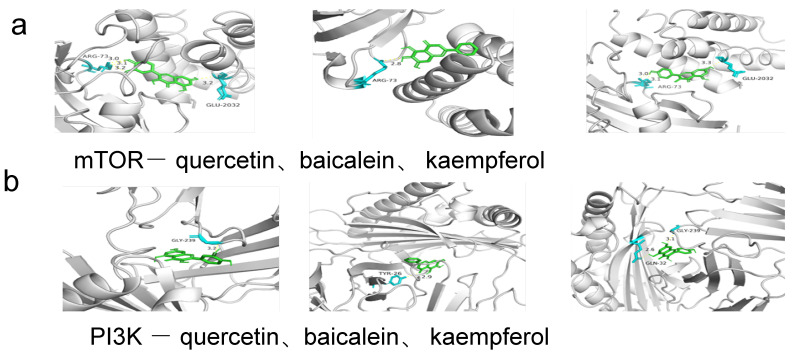
Molecular docking results (note: from left to right: quercetin, baicalein, and kaempferol; (**a**) mTOR (**b**) AKT1 (**c**) PI3K (**d**) STAT3 (**e**) TNF (**f**) IL6.

**Figure 7 cimb-47-00450-f007:**
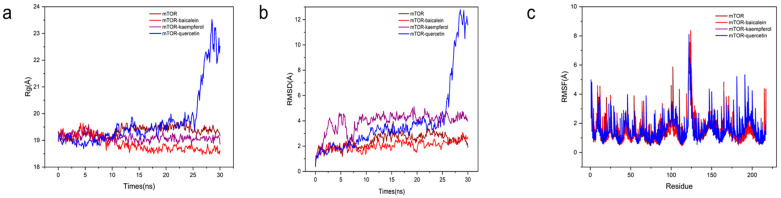
Molecular dynamics simulation results (notes: (**a**) radius of gyration; Rg: represents the protein’s radius of gyration; (**b**) RMSD indicates the root mean square deviation of the protein and ligand; (**c**) RMSF denotes the root mean square fluctuation of amino acid residues).

**Figure 8 cimb-47-00450-f008:**
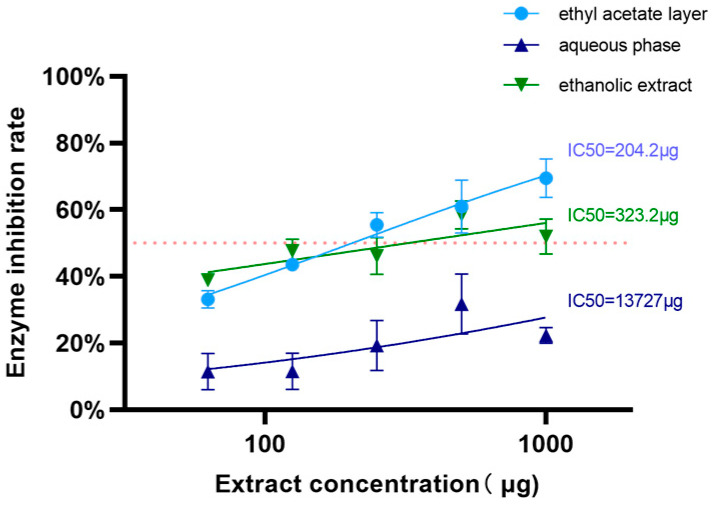
Inhibition rates of pancreatic lipase by different samples.

**Figure 9 cimb-47-00450-f009:**
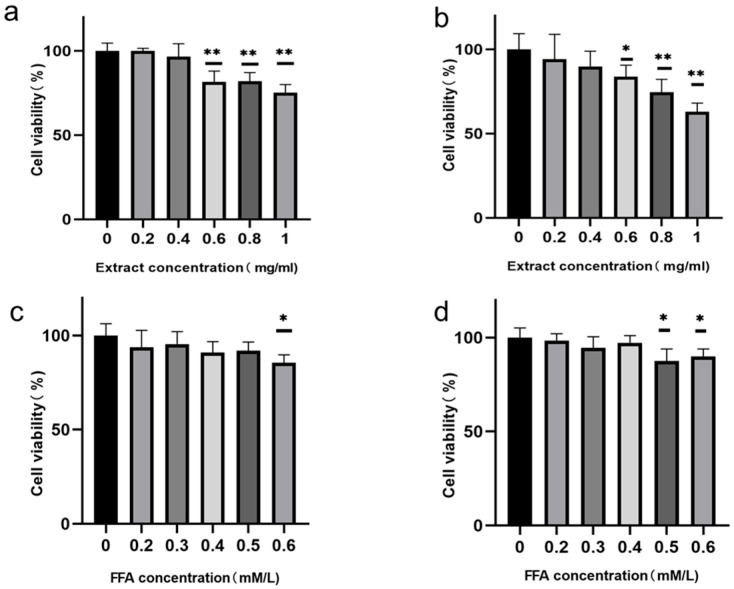
The effects of different concentrations and treatment times of ethyl acetate phase extract and FFAs on cell viability (note: (**a**–**d**) represent the effects of extract and FFAs on cell viability, respectively, with (**a**–**d**) corresponding to treatment times of 24 h and 48 h, respectively; * indicates *p* < 0.05 compared with the 0 concentration group, and ** indicates *p* < 0.01).

**Figure 10 cimb-47-00450-f010:**
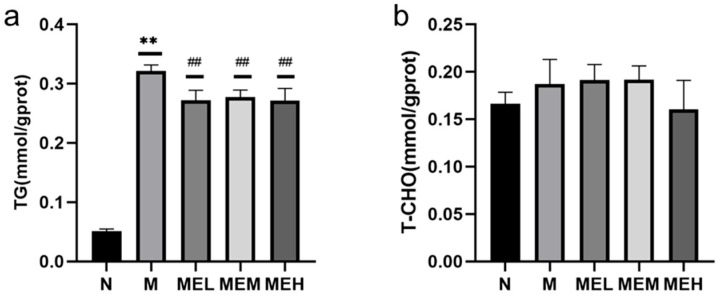
The effects of different concentrations of the extract on TG (triglyceride) and T-CHO (total cholesterol) levels (note: (**a**) represents TG; (**b**) represents T-CHO; ** indicates a significant difference between the model group M and the control group N, with *p* < 0.01; ## indicates a significant difference between the extract group and the model group M, with *p* < 0.01).

**Figure 11 cimb-47-00450-f011:**
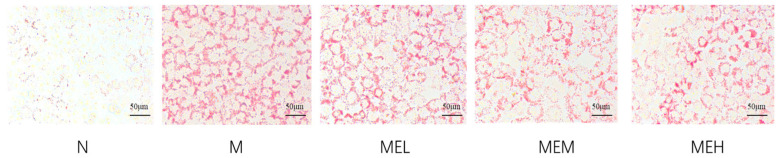
Oil Red O staining results under different groups (note: the observation was made under a 40× microscope; the scale bar represents 50 μm per unit).

**Figure 12 cimb-47-00450-f012:**
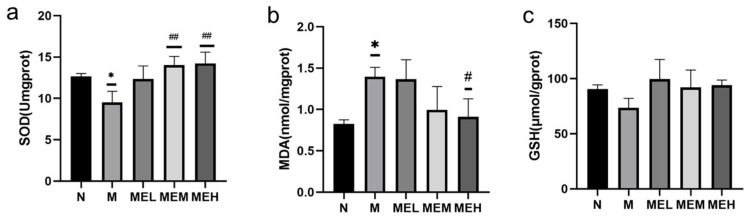
Results of antioxidant indices across different groups (note: (**a**) represents SOD; (**b**) represents MDA; (**c**) represents GSH. * indicate a significant difference between the model group M and the control group N, with *p* < 0.05; # and ## indicate a significant difference between the extract group and the model group M, with *p* < 0.05 and *p* < 0.01, respectively).

**Table 1 cimb-47-00450-t001:** The gradient elution of the mobile phase.

Time (min)	Flow Rate (mL/min)	Phase A (%)	Phase B (%)
0	0.35	95	5
9	0.35	5	95
10	0.35	5	95
11.1	0.35	95	5
14	0.35	95	5

**Table 2 cimb-47-00450-t002:** Mass spectrometry conditions and parameters.

Mass Spectrometry Conditions	Parameters
Source temperature (°C)	500
Ion spray voltage (positive ion mode) (V)	5500
Ion spray voltage (negative ion mode) (V)	4500
Ion source gas I (psi)	50
Ion source gas II (psi)	60
Curtain gas (psi)	25
Collision gas	medium
Collision-activated dissociation	high

**Table 3 cimb-47-00450-t003:** The composition of the reaction system.

Reagents	Sample Group (µL)	Blank Group (µL)	Sample Control Group (µL)	Blank Control Group (µL)
Enzyme solution	50	50	50	50
Sample solution	20	0	20	0
Sample solvent	0	20	0	20
Buffer solution	110			
PNPB	20			

**Table 4 cimb-47-00450-t004:** The OB and DL values of the ingredient compounds.

Compounds	OB (%)	DL
Catechin	49.67	0.24
Skimmin	38.35	0.32
Sesamin	56.55	0.83
Tamarixetin	32.86	0.31
Isorhamnetin	49.60	0.31
Baicalein	33.52	0.21
Fraxin	36.76	0.42
5-O-Coumaroylquinic acid	37.63	0.29
Kaempferol	41.88	0.24
Wogonin	30.68	0.23
Nobiletin	61.67	0.52
Arctiin	34.45	0.84
Gossypetin	35.00	0.31
Laricitrin	35.38	0.34
Azaleatin	54.28	0.30
Syringetin	36.82	0.37
Herbacetin	36.07	0.27
Queretin	46.43	0.28
Morin	46.23	0.27

**Table 5 cimb-47-00450-t005:** Molecular docking results of targets and active components in the PI3K-AKT and insulin resistance pathways.

Ingredient	Receptor	PDB ID	Binding Energy (KJ/mol)	Binding Site
quercetin	PI3K	7I1c	−8.0	GLY-239
quercetin	AKT1	1h10	−6.3	ARG-86, GLU-85, LYS-20
quercetin	IL6	1a1u	−7.1	ARG-179, GLN-175, ARG-30
quercetin	TNF	1a8m	−8.8	SER-99, ARG-103
quercetin	STAT3	5ax3	−8.5	GLN-305, ARG-126, GLU-72
quercetin	mTOR	4dri	−9.2	GLU-2032, ARG-73
baicalein	PI3K	7I1c	−8.0	TYR-26
baicalein	AKT1	1h10	−6.7	HIS-89, GLU-91
baicalein	IL6	1a1u	−6.9	ARG-179, ARG-30
baicalein	TNF	1a8m	−6.5	SER-133, GLN-47, ASP-45
baicalein	STAT3	5ax3	−8.3	ASP-158, GLU-62, GLN-32
baicalein	mTOR	4dri	−9.5	ARG-73
kaempferol	PI3K	7I1c	−7.9	GLY-239, GLN-32
kaempferol	AKT1	1h10	−6.1	ALA-50, LEU-52
kaempferol	IL6	1a1u	−6.7	ARG-179, GLN-175
kaempferol	TNF	1a8m	−6.9	ALA-134, TRP-28
kaempferol	STAT3	5ax3	−8.1	LYS-105, ALA-43
kaempferol	mTOR	4dri	−9.3	ARG-73, GLU-2032

## Data Availability

The original contributions presented in the study are included in the article. Further inquiries can be directed to the corresponding authors.

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
