# Peer review of "Lipid-Lowering Potential of Almond Hulls (Quercetin, Baicalein, and Kaempferol): Insights from Network Pharmacology and Molecular Dynamics"

_cimb, 2025, doi:10.3390/cimb47060450_

Round 1

Reviewer 1 Report

Comments and Suggestions for Authors

Title: Lipid-Lowering Potential of Almond Hulls: Insights from Net- 2 work Pharmacology and Molecular Dynamics (cimb-3638438)

Dear editor,

The topic is interesting. However, there are some questions should be considered.

  1. Title: Please add the key bioactive component of almond hulls in title.
  2. Abstract: The novelty and limitation of this paper should be mentioned.
  3. As much as possible, authors should avoid repeating part of the title words as Keywords as title words can also serve as indicators for searching the manuscript after publication.
  4. Introduction: It is too long. Please provide the Latin name of almond.

Line 54-82: It should be divided into two or more paragraphs. The references should be more specific. Please emphasize the purpose, novelty, and significance of your study.

  1. Methods:

Please provide detailed information of software, instrument model, origin, etc.

  1. Results and discussion

Statistical analysis is insufficient, especially for the data in Table 4, Figure 9, etc.

The discussion often restates results instead of interpreting them in a broader context.

  1. Conclusion is always a hard part of the work to write. It's difficult not just repeat the results and write the conclusions as a resume of the results. But it's always better to make the conclusions a space to draw attention to the relevance of the work and, inclusively, open up possibilities for future studies. What was the contribution that this work brought to the knowledge about experiment? Why is it interesting to do this work? What are the next steps the team hopes to take to continue this research?
  2. Reference: There are double serial number for each reference.
  3. Language should be carefully revised.

Comments on the Quality of English Language
  1. Language should be carefully revised.

Author Response

Comments 1. Title: Please add the key bioactive component of almond hulls in title.

Response 1: Thanks for your suggestion. Following your advice, we have added the relevant content to the title section of the manuscript. Revised the title from Lipid-Lowering Potential of Almond Hulls: Insights from Network Pharmacology and Molecular Dynamics to Lipid-Lowering Potential of Almond Hulls (quercetin, baicalein, and kaempferol): Insights from Network Pharmacology and Molecular Dynamics.

Comments 2. Abstract: The novelty and limitation of this paper should be mentioned..

Response 2: Thanks for your suggestion.Following your advice, the following substantive additions have been incorporated into the abstract section: Network pharmacology and in vitro studies suggest almond hull extract modulates PI3K-AKT signaling and improves insulin resistance, demonstrating lipid-lowering ef-fects. These findings support its potential in functional foods and pharmaceuticals, though further in vivo validation and mechanistic investigations are required.. (line 31-35 )

Comments 3. As much as possible, authors should avoid repeating part of the title words as Keywords as title words can also serve as indicators for searching the manuscript after publication.

Response 3: Thanks for your suggestion. The keyword set has been optimized from “Almond Hull; Network pharmacology; Molecular Dynamics Simulation; HepG2 cells; “to “Almond Hull; Extraction; Mechanisms; HepG2 cells;” to mitigate lexical redundancy within the indexing terms.‌ (line 39)

Comments 4. Introduction: It is too long. Please provide the Latin name of almond.Line 54-82: It should be divided into two or more paragraphs. The references should be more specific. Please emphasize the purpose, novelty, and significance of your study

Response 4 r:.Thanks for your suggestion. The manuscript has been supplemented with the standardized botanical nomenclature (Almond, Amygdalus communis L.) per taxonomic conventions, while the introductory discourse has been strategically streamlined to eliminate discursive redundancy through lexical condensation and systematic annotation of reference citations.‌ (line 68-93)

Comments 5. Methods:Please provide detailed information of software, instrument model, origin, etc.

Response 5: Thank you very much for your questions. Based on the existing Section 2.1 (Materials), we have supplemented additional software-related information throughout the text.‌ (line 249-252)

Comments 6. Results and discussion:Statistical analysis is insufficient, especially for the data in Table 4, Figure 9, etc.The discussion often restates results instead of interpreting them in a broader context.

Response 6: Thanks for your suggestion. Following your advice, we have refined the chart data analysis and augmented the discussion section with substantive revisions. ( line 452-486)‌

Comments 7. Conclusion is always a hard part of the work to write. It's difficult not just repeat the results and write the conclusions as a resume of the results. But it's always better to make the conclusions a space to draw attention to the relevance of the work and, inclusively, open up possibilities for future studies. What was the contribution that this work brought to the knowledge about experiment? Why is it interesting to do this work? What are the next steps the team hopes to take to continue this research?

Response 7: Thanks for your careful checks and patient response. Following your advice, Revised the Conclusion from “This study systematically investigated the hypo-lipidemic mechanisms of Almond hull extract through integrated network pharmacology, molecular docking, dynamics simulations, and in-vitro validation. Key findings revealed that: The extract attenuated lipid accumulation by suppressing pancreatic lipase activity (IC50: 204.2 µg/mL) and reducing intracellular triglyceride (TG) deposition in HepG2 cells (p < 0.05 at 0.1 mg/mL), while concurrently enhancing antioxidant defenses via SOD restoration and MDA/GSH modulation. Network pharmacology and molecular docking identified polyphenolic constituents (quercetin, baicalein, kaempferol) as putative bioactive agents targeting the PI3K-AKT signaling axis and insulin resistance pathways, potentially regulating lipid metabolism and inflammatory cascades. These results provide mechanistic evidence for the extract's dual-phase lipid-lowering effects, positioning Almond hulls as a sustainable resource for developing multifunctional nutraceuticals or cosmeceuticals targeting metabolic syndrome” to “This study comprehensively elucidated the hypolipidemic mechanisms of almond hull extract through an integrated approach combining network pharmacology, molecular docking, dynamic simulations, and in vitro validation. Key findings demonstrated significant lipid-lowering effects through pancreatic lipase inhibition (in vitro assay) and reduced intracellular triglyceride accumulation (HepG2 cell model), ac-companied by enhanced antioxidant capacity. Our results not only confirm the ex-tract's anti-obesity potential but also identify critical signaling pathways for future mechanistic validation. These findings establish a theoretical foundation for develop-ing almond hull-derived nutraceuticals targeting metabolic syndrome. Future studies will focus on compound purification and in vivo/in vitro mechanistic investigations to further validate these pathways”.

Comments 8. Reference: There are double serial number for each reference.- Figure 5 and figure 6 are impossible to understand.

Response 8: We appreciate your identification of this oversight and have accordingly updated the reference numbering.‌

Comments 9.  Language should be carefully revised.

Response 9: We thank you for highlighting these concerns and have accordingly polished the linguistic elements throughout the manuscript.‌

Reviewer 2 Report

Comments and Suggestions for Authors

Dear Editor and Authors,

The manuscript ‘Lipid-Lowering Potential of Almond Hulls: Insights from Network Pharmacology and Molecular Dynamics’ by Qiming Miao 1, Lu Sun 1, Jiayuan Wu 1,Xinyue Zhu 1, Juer Liu 2, Roger Ruan 2, Guangwei Huang 3, Shengquan 4 Mi 1, * and Yanling Cheng 1 is a research study on almond hulls extracts using network pharmacology and molecular docking to investigate the interactions between key bioactive constituents (e.g., quercetin, 24 baicalein, and kaempferol) and targets in lipid metabolism-related pathways.

The Authors performed molecular docking and in vitro assays such as pancreatic lipase inhibition assay, cellular lipid-lowering experiment, quantitative analysis of cellular triglycerides, total cholesterol, and antioxi dant biomarkers, oil red O staining.

Based on the results the Authors stated that the extract attenuated lipid accumulation by suppressing pancreatic lipase activity and reducing intracellular triglyceride (TG) deposition in HepG2 cells, while concurrently enhancing antioxidant defenses via SOD restoration and MDA/GSH modulation. Network pharmacology and molecular docking identified polyphenolic constituents (quercetin, baicalein, kaempferol) as putative bioactive agents targeting the PI3K-AKT signaling axis and insulin resistance pathways, potentially regulating lipid metabolism and inflammatory cascades. The results were obtained in vitro.

The manuscript is well written and interesting. The Authors collected 53 references on the topic.

However, there is lack of quantitative composition of the extracts.

In Figure 10. Caption abbreviations MEL/MEM/MEH should be explained

Cordially

Author Response

Comments 1 there is lack of quantitative composition of the extracts.

Response 1: We appreciate your endorsement of our work. Given our primary research focus on qualitative phytochemical profiling of the extracts, quantitative validation of core bioactive constituents derived from network pharmacology remains beyond the current scope. We will prioritize this analytical extension in subsequent investigations as per your valuable recommendation.‌

Comments 2 Figure 10. Caption abbreviations MEL/MEM/MEH should be explained

Response 2: Thank you very much for your suggestion. The experimental designations MEL/MEM/MEH were explicitly defined in Section 2.4.2.2 (Cellular Lipid Reduction Assay Protocol). We acknowledge that the manuscript's structural organization may have impeded textual navigation, and sincerely appreciate your rigorous examination.‌